# Repeated examples help learn arithmetic

**François Charton**
FAIR, Meta & Ecole des Ponts
fcharton@meta.com

**Julia Kempe**
FAIR, Meta & NYU CDS and Courant Institute
kempe@meta.com

## Abstract

We study small transformers trained on two problems of arithmetic: the greatest common divisor (GCD) and modular multiplication, and show that models trained on a limited set of repeated examples achieve better performance than models trained from unlimited data. In fact, modular multiplication is only learned on small training sets. We also demonstrate that *two-set training* - repeated use of a small random subset of examples, along normal sampling on the rest of the training set - provides for faster learning and better performance. These experiments highlight that the benefits of repetition can outweigh those of data diversity; and shed light on the still poorly understood interplay between generalization and memorization in deep learning.

## 1 Introduction

When training neural networks, it has become customary to use the largest and most diverse datasets available, and to limit example reuse as much as possible. On problems of arithmetic, the training data is easy to generate in very large quantities, sometimes even on the fly. For this reason, models are trained on very large sets of single-use examples, with very low repetition. In this paper, we investigate the *low data regime*: transformers trained on smaller sets of repeated examples. We consider two problems: the greatest common divisor (GCD, Charton (2024)) of two integers, and multiplication modulo 67 of two positive integers between 1 and a million, and measure model performance for different *data budgets* (DB, the number of distinct examples in the training set), and *training budgets* (TB, the total number of training examples). On the GCD problem, we show that, for a given training budget, models trained on small data budgets (but large enough to avoid overfitting) outperform models trained on large or unlimited data budgets. We also show that modular multiplication is *only learned* for small data budgets. These experiments demonstrate the benefit of training on repeated examples, challenging the common idea that one or two epochs is all we need.

Pushing this observation further, we demonstrate that for a given data budget, model performance can be greatly improved by *two-set training*: selecting at random a small subset of training examples, and repeating them more often during training. This two-set effect is all the more surprising as the repeated examples are not curated, and only differ from the rest of the training sample by their frequency of reuse. In fact, ablation experiments indicate that the performance of two-set training cannot be improved by curating the set of repeated examples, or refreshing it as training proceeds. We also show that mixing repeated and non-repeated examples in the same mini-batches is a necessary step for the two-set effect to appear.

The benefits of repetition are significant in both problems, but come in different flavors. For GCD, repetition allows for better performance and faster learning. For modular multiplication, it unlocks an emergent capability: without repetition, the model does not learn. We believe these findings have profound implications and should lead to a paradigm shift where the training set size becomes a mere hyper-parameter, not solely governed by the availability of data and the belief that more is always better.

38th Conference on Neural Information Processing Systems (NeurIPS 2024).

## 2   Experimental settings

We focus on two problems of arithmetic: computing GCD and multiplication modulo 67. The GCD was studied in prior work (Charton, 2024; Dohmatob et al., 2024).

In the **greatest common divisor** problem, the model is tasked to predict the GCD of two integers uniformly distributed between 1 and 1 million, encoded in base 1000. Following Charton (2024), who observes that throughout training almost all pairs of integers with the same GCD are predicted the same, we evaluate model performance by the number of GCD below 100 predicted correctly, measured on a random test sample of $100,000$ pairs: 1000 pairs for each GCD from 1 to 100. On this metric, Charton (2024) reports a best performance of 22 correct GCD for a model trained on uniformly distributed inputs. This test metric is preferable to a more standard measure of accuracy on random input pairs, because GCD are distributed according to an inverse square law ($61\%$ of random pairs have GCD 1, $15\%$ have GCD 2), making accuracy a very optimistic measure of performance.

In **modular multiplication**, we train models to predict the product, modulo 67, of two integers between 1 and a million. Arithmetic modulo $p$ was studied in several previous works, in the context of grokking (Power et al., 2022; Liu et al., 2022a) and mechanistic interpretability (Zhong et al., 2023), but with model inputs sampled from 0 to $p - 1$, which results in a very small problem space for small $p$. We evaluate model accuracy as the percentage of correct predictions of $a \times b \mod 67$, on a test set of $10,000$ examples (generated afresh at every evaluation).

**Models and tokenizers.** We use sequence-to-sequence transformers (Vaswani et al., 2017) with 4 layers in the encoder and decoder, an embedding dimension of 512, and 8 attention heads (35 million trainable parameters). Models are trained to minimize a cross-entropy loss, using the Adam optimizer (Kingma & Ba, 2014), with a learning rate of $10^{-5}$, and batches of 64. The integer inputs and outputs of both problems are tokenized as sequences of digits in base 1000, preceded by a sign which serves as a separator. All experiments are run on one V100 GPU with 32 GB of memory.

## 3   Repetition Helps

In a first series of experiments, we compare the performances of models trained on different data budgets (number of distinct examples) for increasing training budgets (total examples). We consider DB of $1, 5, 10, 25, 50$ and 100 million examples, together with the unlimited case, where examples are generated on the fly (yielding DB≈TB). Figure 1 (Left) presents the average number of GCD predicted by 5 models trained on different DB, for increasing TB. For a small TB of 30 million examples, models trained on 1 and 5M DB achieve the best performance: 20 GCD vs 13 for all others. As TB increases, 1M-models start overfitting, as shown by increasing test losses in Figure 1 (Right), and their performance saturates at 21 correct GCD. The performance of the 5M models keeps improving to 36 GCD, for a TB of 150 million examples, then begins to overfit, and saturates around 38. For TB of 150 and 300 million examples, the best performing models are the 10M.

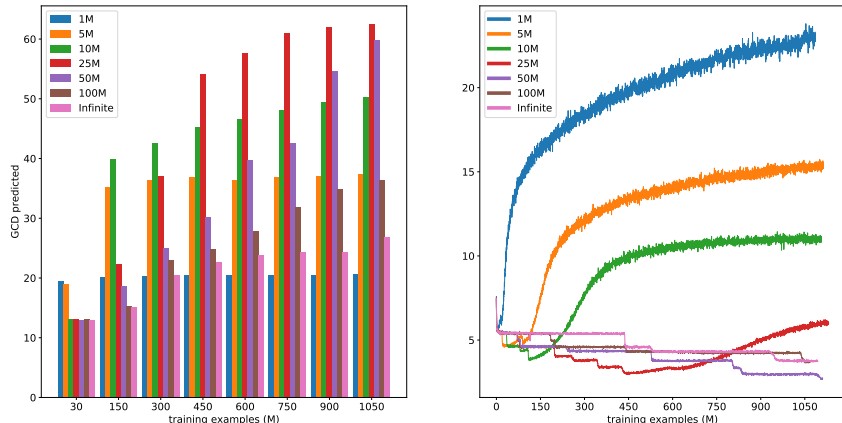

Figure 1: **GCD problem:** (Left) GCD accuracy for different data and training budgets (average of 5 models). (Right) Test loss of models as a function of training budget, for fixed data budgets.

For TB between 450M and 1.05 billion examples, models trained on 25M DB achieve the best performances. Throughout training, the models trained on small data budgets learn fastest. Past a certain TB, they overfit their training data, and their performance saturates. On the other hand, models trained on large or unlimited DB perform the worst. For a TB of one billion examples, models trained on 100M or unlimited data budgets only predict 37 and 27 GCD, way worse than models trained on 25 and 50M (62 and 60). When learning GCD, smaller data budgets and more frequent repetition allow for faster learning, and much better performance.

**Modular multiplication**, with a TB of 600 million, tells a different story. Models with DB of 10M or less, or 100M or more do not learn the task, and achieve about chance level accuracies, predicting all outcomes as 0, for an accuracy slightly over 3% (Table 1). On the other hand, models trained on 25 and 50M distinct examples (repeated 24 and 12 times on average) do learn the task: 25% of models trained with this DB achieve 99% accuracy, and a majority achieves 50%. On this task, learning emerges from repetition: models trained on small DB learn a task inaccessible to models trained on larger DB. (note: increasing TB to 2B examples, some models trained on 100MDB do learn, but none of the unlimited data models do).

Table 1: **Multiplication modulo 67**. Accuracy of models trained on a budget of 600 million data points.

| | Data budget (millions) | | | | | | |
| --- | --- | --- | --- | --- | --- | --- | --- |
| | 1 | 5 | 10 | 25 | 50 | 100 | unlimited |
| Average accuracy (%) | 1.6 | 3.8 | 4.4 | 40.4 | **59.5** | 5.4 | 3.0 |
| Number of models achieving 99% accuracy | 0/5 | 0/5 | 0/5 | 6/25 | **7/25** | 0/30 | 0/30 |
| Number of models achieving 50%+ accuracy | 0/5 | 0/5 | 0/5 | 13/25 | **22/25** | 0/30 | 0/30 |
| Number of models trained | 5 | 5 | 5 | 25 | 25 | 30 | 30 |

These experiments clearly indicate that repetition helps learning. On both tasks, for a fixed training budget, models trained on a small data budget, i.e. fewer distinct examples, repeated several times, achieve much better performance than models trained from single-use examples, or repeated very few times, as is customary in most recent works on language models (Muennighoff et al., 2023). Smaller data budgets and repeated examples elicit "emergent learning".

## 4   Two-set training

We now turn to a different problem: how to best use a given data budget? Because repetition helps learning, we want a small subset of repeated examples, but we also observed that, after a certain training budget, models trained on small datasets start overfitting, which cause their performance to saturate. To balance these two effects, we propose **two-set training**: randomly splitting the training sample into a small set of $S$ examples, used with probability $p$, and repeated many times, and a large set, used with probability $1 - p$, and repeated a few times. By doing so, we hope that the small set fosters learning, while the large set prevents overfitting.

On the **GCD problem**, with a data budget of 100 million examples and a training budget of 600 million, models trained on a repeated set of 250,000 examples or less, used with a probability of $p$ of 0.25 or 0.5, predict more than 62 GCD on average (Figure 2), vs 27 in the single-set case. The best results, 69 GCD, are achieved for $S = 50,000$ and $p = 0.25$. In this setting, small set examples are repeated 3000 times on average, large set examples 4.5 times.

These results extend to unlimited data budgets, using a fixed set of $S$ examples. The best choices of $p$ and $S$ are roughly the same as with a DB of 100M (Figure 3 in Appendix B). For $p = 0.25$ and $S = 50,000$, two-set training achieves an average performance of 67 GCD on 6 models, a spectacular improvement over models trained on a single set, which predict 25 GCD on average. For smaller DB (25 or 50M), two-set training provides for faster learning (Figure 4, Appendix B).

For **modular multiplication**, we need larger samples of repeated examples: $S$ between 2.5 and 10M examples for 100M DB, and $S = 25$M with unlimited DB (Figure 5 Appendix C). Examples from the small set are repeated less: from 20 to 60 times, vs 3000 for the GCD. Yet, two-set training results in improved performances for all data budgets over 25M (Table 2). More than 50% of all the models we trained could learn modular multiplication with over 99% accuracy (90% with more than 50% for DB larger than 50M). In contrast, single-set training achieved 99% accuracy for 25% of models with DB 50M or less, and for none of the models trained on larger data budgets (Table 1).

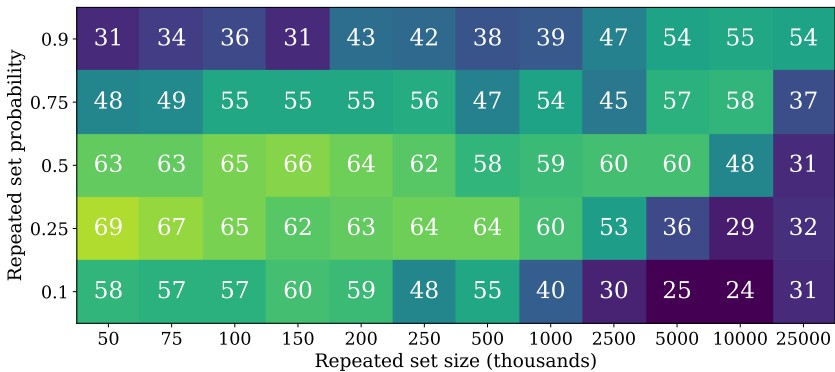

Figure 2: **Two-set training for the GCD problem:** Number of correctly predicted GCD as a function of $S$ and $p$. Each measurement is the average of 6 models. Data budget 100M, training budget 600M. Note the high performance for very small sets $S$ of sizes 50, 75, 100, 150 and 200 thousand, with $p = 0.25$ and $p = 0.5$.

Table 2: **Two-set training on modular multiplication.** Percentage of models (different random initializations) learning to compute modular multiplication with 50 and 99% accuracy. Training budget: 600M. For DB 25M and 50M, 10 models with two-set training, and 25 with single set training. For DB 100M and unlimited, 26 models with two-set training, and 30 with single set training.

| Data budget | p / S | Two sets | | Single set | |
|---|---|---|---|---|---|
| | | $> 50\%$ | $> 99\%$ | $> 50\%$ | $> 99\%$ |
| 25M | 0.1 / 1M | 50 | **50** | 52 | 24 |
| 50M | 0.25 / 2.5M | 90 | **50** | 88 | 28 |
| 100M | 0.5 / 10M | 88 | **54** | 0 | 0 |
| Unlimited | 0.25 / 2.5M | 92 | **58** | 0 | 0 |

Ablation studies (Appendix D) indicate that curating the small sample – selecting easier, or particular examples for the repeated set (an obvious strategy for improving two-set training), brings at best a marginal increase in performance. They also indicate that mixing repeated and non-repeated examples in the same mini-batches is an essential element of two-set training. Models trained on batches exclusively selected from the small and large sets do not learn.

## 5    Conclusion

Our findings indicate that the performance of math transformers can be greatly improved by training them on datasets which include repeated examples. This can be done by using smaller train set, or randomly selecting a repeated subset from a larger corpus. On the GCD problem, repeated examples allow for faster learning and better performance. For modular arithmetic, they are necessary for the model to learn. This suggests that abandoning the customary practice of training models on the largest possible set of single-use example may be beneficial.

Our observations on two-set training, in particular the fact that the repeated set can be selected at random (and that curation does not help) are thought-provoking. All that seems to matter is that the *exact same* examples are repeated, i.e. not their informational content. This is all the more shocking as repetition occurs at a very low frequency. In the GCD experiments, examples in the small set are repeated 3000 times over a TB of 600 millions: once in 200,000 examples on average. For modular multiplication, the frequency is even lower. Besides, the repeated examples *need to be mixed* with non-repeated examples into mini-batches for the two-set effect to appear.

This raises several tantalizing questions: how does the transformer "figure" that a given example, lost in a minibatch, has been seen, hundreds of thousands of examples before? Our research suggests that there exists a qualitative difference between "déjà vu" and "jamais vu" examples – data points the model has already seen, or never seen. How do transformers, and perhaps other architectures, identify, and then process, "déjà vu" examples? To our knowledge, this aspect was overlooked in many prior works on model interpretation. It is an intriguing subject for further study.

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

## Appendix

## A  Background and Related Work

In this paper, we focus on relatively small transformer models performing mathematical tasks, placing it into a long established corpus of works that study interesting phenomena in a controlled setting, and advance our understanding of the underlying mechanisms in larger models in the wild, see e.g. Power et al. (2022); Garg et al. (2022); Charton (2024); Dohmatob et al. (2024).

One such example is the study of *"grokking"*, first observed with modular arithmetic - a phenomenon where models generalize long after achieving $100\%$ accuracy on their (small) training set (Power et al., 2022; Liu et al., 2022b, 2023). On the surface, grokking shares similarities with our work: a small training dataset is iterated for many epochs, the phenomenon is isolated in clean experiments on synthetic data, and it contradicts traditional wisdom regarding overfitting (Mohri et al., 2018). But there are important differences: in grokking, delayed learning occurs, we observe no such delay; grokking occurs for "tiny" training samples (hundreds or thousands of examples), our models use millions (even for modular multiplication); grokking is very sensitive to the optimizer used, our findings are robust across optimizers (Appendix D.5), and, of course, no two-set approach is documented in the grokking setting.

Another related setting is *"benign overfitting"* (Bartlett et al., 2020; Belkin, 2021; Bartlett et al., 2021), where an *over-parametrized* model perfectly fits noisy data, without harming prediction accuracy. One could argue that our work presents a *quantitative* manifestation of benign overfitting, inasmuch as decreasing the data budget increases model over-parametrization. However, this would not account for the decrease in performance once the data budget falls below a certain number (one could argue that overfitting is no longer benign, then), nor for the possibility of two-set training.

Our work is related to, but different from, *curriculum learning (CL)* (Bengio et al., 2009; Wang et al., 2022), where training data is presented in a meaningful order, usually from "easy" to "hard" samples. Two-set training, differs from curriculum learning in at least two important ways: in CL, datasets are curated, our subsets are completely random; in CL, the training distribution shifts over time, while our subsets are static. Our ablations show that curating the repeated set, or changing it over time, as in CL, brings no improvement on performance (and may even have an adverse effect).

Lastly, our work touches upon the expansive area of *out-of-distribution (OOD)* generalization (Gulrajani & Lopez-Paz, 2021; Lopez-Paz, 2025), which studies generalization when train and test distributions differ. Curiously, while our two-set approach increases the frequency of some training examples, because the repeated set is chosen *at random*, the training set remains distributionally equivalent to the test set. Thus, our study falls outside the usual framework of OOD studies.

# B  Additional figures for GCD experiments

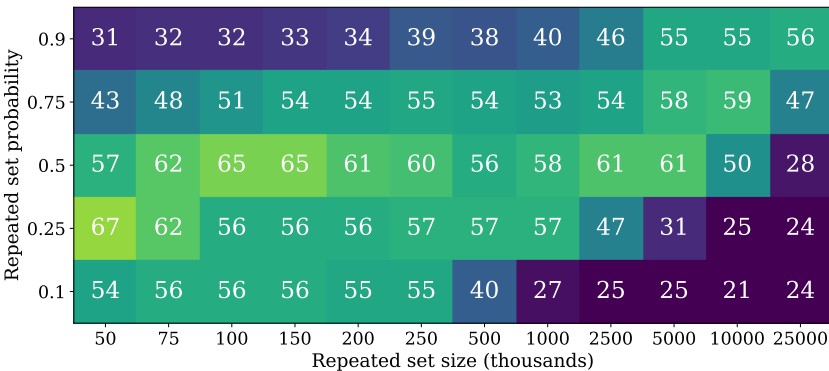

Figure 3: **Two-sample training for the GCD problem for $\infty$-models:** Number of correctly predicted GCD as a function of small set size $S$ and $p$, each averaged over 6 models. Data budget *and* training budget equal 600M ($\infty$-models). Note the high performance for very small sets $S$ of sizes between 50 and 200 thousand, with $p = 0.25$ and $p = 0.5$ compared to "standard" training with the same data budget, predicting 25 GCD.).

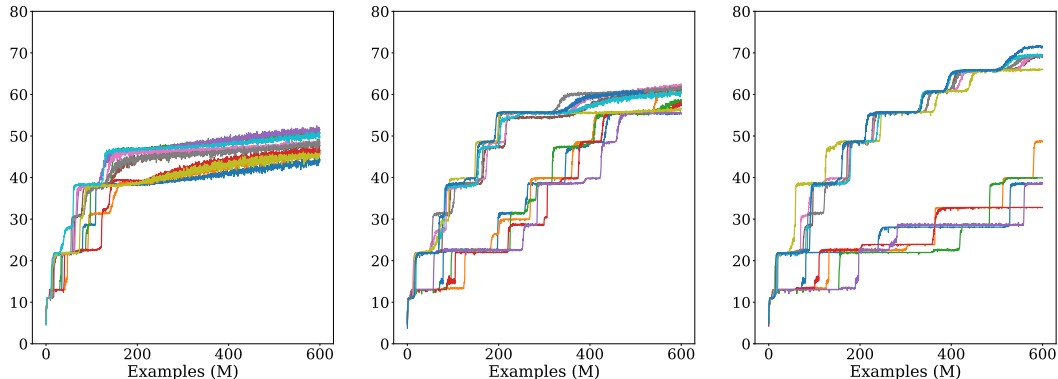

Figure 4: **Two-set versus single-set training for the GCD problem:** Number of correctly predicted (test) GCD as a function of training budget (up to 600M) for data budgets of 10M (left), 25M (center), and 50M (right). Two-set training with $p = 0.25$ and $|S| = 50,000$ (top 6 curves) versus single-set training (lower 6 curves).

## C   Additional figures for modular multiplication

Figures 5 as well as Tables 3 and 4 provide additional results for modular multiplication in the two-set setting.

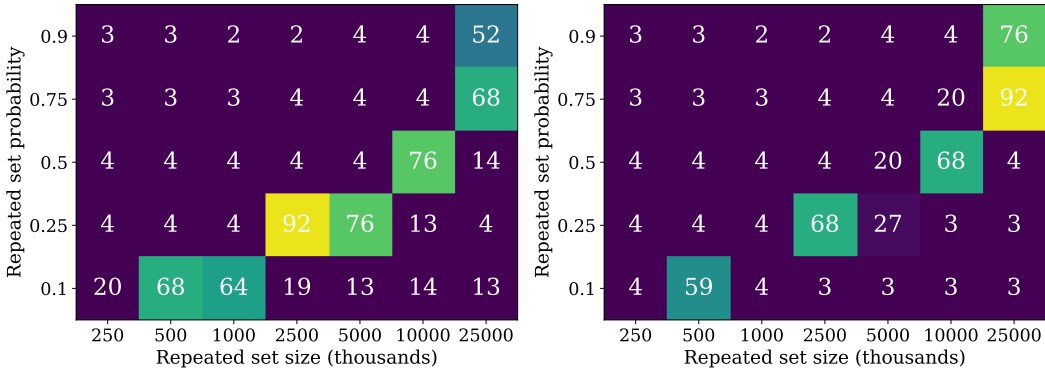

Figure 5: **Two-set training for Modular Multiplication:** Accuracy as a function of small set size $S$ and $p$, each averaged over 6 models. Data budget 100M (left) and unlimited (right), training budget 600M. Note: the bottom right of the left graph correspond to single-set 10M-models: for $p = 0.1$ and $S = 10$M, the small and large set are selected with the same probability.

Table 3: **Two-set training on modular multiplication.** For a training budget of 600M we show the number of models (random initializations) that achieve $50 + \%$ and $90\%$ accuracy for several data budgets and sizes of the more frequent sets $S$, and probabilities $p$. The baseline of single-set traning from Section 3 is given in the last line. Similar results for training budgets of 300M and 450M are given in Table 4.

| $(p, S)$/ **Data budget** | 25**M** | | 50**M** | | 100**M** | | $\infty$ | |
|---|---|---|---|---|---|---|---|---|
| | $> 50\%$ | $99\%$ | $> 50\%$ | $99\%$ | $> 50\%$ | $99\%$ | $> 50\%$ | $99\%$ |
| $(0.1, 500K)$ | 2/10 | 1/10 | 6/10 | 3/10 | 20/26 | 10/26 | **25/26** | 8/26 |
| $(0.1, 1M)$ | **5/10** | **5/10** | 8/10 | 4/10 | 22/26 | 6/26 | 0/26 | 0/26 |
| $(0.25, 2.5M)$ | 2/10 | 1/10 | **9/10** | **5/10** | 20/26 | 9/26 | 24/26 | **15/26** |
| $(0.25, 5M)$ | 3/10 | 1/10 | 9/10 | 4/10 | **24/26** | 10/26 | 5/26 | 0/26 |
| $(0.5, 10M)$ | 3/10 | 3/10 | 8/10 | 5/10 | 23/26 | **14/26** | 23/26 | 12/26 |
| $(0.75, 25M)$ | - | - | - | - | 23/26 | 10/26 | 20/26 | 14/26 |
| Single set | 13/25 | 6/25 | 22/25 | 7/25 | 0/30 | 0/30 | 0/30 | 0/30 |

Table 4: **Two-set training on modular multiplication.** For training budgets of 300M, 450M and 600M we show the number of models out of 10 (random initializations) that achieve $50 + \%$ and $90\%$ accuracy for data budgets 25M and 50M, and sizes of the more frequent sets $S$, and probabilities $p$. The baseline of single-set training is given in the last line, out of 25 models. The next to last line renormalizes this to out of 10.

| | **Data budget** 25**M** | | | | | | **Data budget** 50**M** | | | | | |
|---|---|---|---|---|---|---|---|---|---|---|---|---|
| | $> 50\%$ | | | $99\%$ | | | $> 50\%$ | | | $99\%$ | | |
| | 300M | 450M | 600M | 300M | 450M | 600M | 300M | 450M | 600M | 300M | 450M | 600M |
| $(0.1, 500K)$ | 1 | 2 | 2 | 0 | 1 | 1 | 4 | 5 | 6 | 0 | 1 | 3 |
| $(0.1, 1M)$ | 1 | **5** | **5** | 0 | **3** | **5** | 3 | 6 | 8 | 0 | 1 | 4 |
| $(0.25, 2.5M)$ | 2 | 2 | 2 | 0 | 1 | 1 | 5 | **9** | **9** | 0 | 1 | **5** |
| $(0.25, 5M)$ | **3** | 3 | 3 | 0 | 0 | 1 | 4 | **9** | **9** | 0 | 1 | 4 |
| $(0.5, 10M)$ | 2 | 3 | 3 | 0 | 2 | 3 | **7** | 7 | 8 | 0 | **2** | 5 |
| Single set (/10) | 3.6 | 4.8 | 5.2 | 0.4 | 1.2 | 2.4 | 2.4 | 7.6 | 8.8 | 0 | 0.8 | 2.8 |
| Single set (/25) | 9/25 | 12/25 | 13/25 | 1/25 | 3/25 | 6/25 | 6/25 | 19/25 | 22/25 | 0/25 | 2/25 | 7/25 |

# D Ablation results

## D.1 Curating the small sample

In two-set training, the examples in the small set are chosen at random from the overall training set. In this section, we experiment with curating the small set, by *selecting* the examples that will be repeated during training. As in curriculum learning, selecting easier or more informative examples may help improve performance. Perhaps when increasing the frequency of our small random set, what really matters is the repetition of some particular examples, rather than all? The GCD problem is particularly well suited for this type of investigation, due to the inverse polynomial distribution of outcomes ($\mathrm{Prob}(\mathrm{GCD} = k) \sim \frac{1}{k^2}$). On this problem, we leverage the findings of Charton (2024), who observes that $\infty$-models trained from log-uniform distributions of inputs and/or outcomes ($\mathrm{Prob}(\mathrm{GCD} = k) \sim \frac{1}{k}$) learn better.

We experiment with four settings of $|S|$ and $p$, which correspond to the best results in our previous experiments (Section 4): $50,000$ and $150,000$ with $p = 0.25$ and $150,000$ and $500,000$ with $p = 0.5$, for a data budget of 100 million and training budget of 600M. For every setting, we train 5 models with the following three choices for $S$: log-uniform inputs, uniform GCD or both log-uniform inputs and GCD. We use two-sample training with a random small set $S$ as our baseline. Table 5 shows that the performance of models using log-uniform inputs, or uniform GCD, is slightly lower than the baseline. Models trained on log-uniform inputs and GCD achieve slightly better performance, but we note that models trained on the small set distribution only ($p = 1$) would predict 91 GCD . On these three distributions, curating the small set proves disappointing.

In curriculum learning fashion, we also experiment with small sets $S$ of a few "easier cases": small inputs (from 1 to 1000), GCD that are products of 2 and 5, the easiest to learn in base 1000 (Charton, 2024), and GCD between 1 and 10 (the most common outcomes). We observe that while models trained with small inputs in $S$ perform on par with the baseline, models trained on "easy GCD" perform slightly worse.

Finally, inspired by arguments that rare tail outcomes might require particular attention for learning (Dohmatob et al., 2024), we experiment with small sets composed of examples from the tail of the training distribution, namely, large GCD. Charton (2024) observes that these are both harder to learn, and less common in the training set. Specifically, we create $S$ with examples with GCD larger than $k$ (for $k$ ranging from 1 to 5). While experiments achieve the best accuracies compared to the other curation schemes we proposed, and values of $k$ equal to 2 and 3 train slightly faster, they remain a little below the baseline both in accuracy and learning speed.

Table 5: **GCD problem: cherry-picking the small set**. (Left) Number of (test) GCD predicted for training budget of 600 million examples, average of 5 models (3 models for baseline). **bold**: more than 65 GCD predicted. (Right) Training budget needed to predict 60 GCD, fastest of 20 models (of 12 models for baseline).

| | 50k / 0.25 | 150k / 0.25 | 150k / 0.5 | 500K / 0.5 | Training budget for 60 GCD (M) |
|---|---|---|---|---|---|
| Log-uniform inputs | 55.9 | 59.4 | 57.9 | 62.0 | 332 |
| Uniform GCD | 55.9 | 54.5 | 41.9 | 54.9 | - |
| Log-uniform inputs and GCD | 62.2 | **71.7** | **66.5** | **72.6** | 88 |
| Small inputs (1-1000) | 61.2 | **67.5** | 62.6 | 62.9 | 247 |
| GCD 1- 10 | 59.9 | 63.8 | 55.8 | 62.3 | 401 |
| GCD products of 2 and 5 | 54.2 | 39.8 | 40.7 | 30.1 | 548 |
| All GCD but 1 | **65.4** | 63.7 | 56.7 | 58.1 | 405 |
| All GCD but 1,2 | **66.8** | 60.0 | 62.8 | 56.9 | 326 |
| All GCD but 1,2,3 | **66.7** | 58.4 | 62.8 | 58.2 | 327 |
| All GCD but 1,2,3,4 | **65.5** | 60.3 | 62.8 | 56.9 | 379 |
| All GCD but 1,2,3,4,5 | **66.5** | 60.6 | 64.9 | 56.3 | 376 |
| GCD product of 2, 3, and 5 | **66.1** | 59.4 | 59.8 | 47.3 | 359 |
| Prime GCD | 64.9 | 62.5 | 58.8 | 64.7 | 422 |
| GCD divisible by primes $\geq 11$ | 60.1 | 54.4 | 35.7 | 42.7 | 569 |
| Baseline (two-set training) | **69.4** | 61.9 | **65.9** | 59.4 | 373 |

Overall, these experiments suggest that in two-set training, random selection of the small set may be optimal. Selecting a small set of easy cases (GCD multiple of 2 and 5), and examples that are known to help training (log-uniform inputs) does not help, and limiting the small set to edge cases from the tail of the outcome distribution brings no improvement to performance. This is a counter-intuitive, but significant result.

## D.2 Batching in two-set training: mixed batches are needed

In all experiments, during training, the model computes gradients over minibatches of 64 examples. In two-set training, minibatches mix examples from the small and large set. We experimented with using "mono-batches" that use samples from one set at a time. For instance, when training with $p = 0.25$, 25% of minibatches would use examples from $S$ only, and 75% would only use those from $\overline{S}$.

On the **GCD problem**, we rerun the most successful two-set experiments (Section 4) with "mono-batches" for $S = 50K$, 100K and 250K, and $p = 0.25$ and 0.5. For training budgets of 600M and data budget of 100M examples, the models trained on mixed batches predicted 62 to 69 GCD (Section 4). With "mono-batches", the number of correctly predicted GCD never rises above 15. For **modular multiplication**, we experimented with the following $(S, p)$ pairs ($S$ in millions): $(0.5, 0.1), (2.5, 0.25)$ and $(10, 0.5)$ with data budget 100M and training budget 600M. With these settings, mixed-batch models achieve an average accuracy of 67% or more (Section 4). With "mono-batches", none of the models manages to learn (accuracy around 4%). This indicates that **mixed batching of samples from each of the two sets plays a central role for the two-set effect**.

## D.3 Shifting the small set

In these experiments, we study, in two-set training, the possible impact of overfitting on the small set, by refreshing the small set with fresh examples periodically. This mimics certain aspects of curriculum learning, where the training set is changed over time. On the GCD experiments, with a data budget of 100 million, a training budget of 600 million, we shift the small set as training proceeds, so that examples in the small set are seen $k$ times on average. At the beginning of training, the small set is the $S$ first elements in the train set. After training on $kS/p$ examples, examples in the small set have been seen $k$ times, and the small set is shifted to elements $S + 1$ to $2S$ of the training set.

Table 6 provides performances for two-set training with shift, for different values of $p$, $S$ and $k$, for a data budget of 100 million, and a training budget of 600 million. It is interesting to note that shifting brings no improvement to 2-set training.

Table 6: **Shifted two-set training.** GCD predicted, average of 3 models, trained on a budget of 600 millions, and a data budget of 100 million, for different values of S, p and k.

| $S$ | 250,000 | | | | 500,000 | | | | 1,000,000 | | | |
| k | 10 | 25 | 50 | 100 | 10 | 25 | 50 | 100 | 10 | 25 | 50 | 100 |
|---|---|---|---|---|---|---|---|---|---|---|---|---|
| $p = 1.0$ | 37 | 22 | 21 | 22 | 37 | 38 | 30 | 31 | 55 | 45 | 37 | 30 |
| $p = 0.9$ | 47 | 38 | 38 | 38 | 55 | 47 | 43 | 39 | 55 | 48 | 47 | 47 |
| $p = 0.75$ | 56 | 38 | 54 | 48 | 56 | 55 | 49 | 55 | 60 | 56 | 55 | 56 |
| $p = 0.5$ | 61 | 56 | 56 | 58 | 61 | 60 | 56 | 58 | 64 | 63 | 63 | 61 |
| $p = 0.25$ | 56 | 62 | 61 | 63 | 49 | 63 | 63 | 61 | 49 | 63 | 62 | 63 |

## D.4 From two-set to many-set training

Two-set training with a small randomly selected subset $S$ amounts to assigning different probabilities to elements in the training set. For a randomly shuffled training set of size $N$, two-set training amounts to selecting the first $S$ elements with probability $p/S$ (with replacement) and the $N - S$ last with probability $(1-p)/(N-S)$, a step-function distribution over $\{1, \ldots, N\}$. We now generalize this approach by introducing a probability law $P$ such that $P(i)$ is the probability of selecting the $i$-th example in the training set. Our motivation is to obtain a smooth, possibly more principled, distribution than the step-function induced by the two-set approach. Pragmatically, a one-parameter

family of smooth distributions eliminates the need to tune both $S$ and $p$. Lastly, we can study whether a smooth decay in frequency might be even more beneficial than a non-continuous two-set partition.

In this section, we consider a discrete exponential distribution:

$$P(i) \sim \beta e^{-\beta i/N},$$

with $\beta > 0$, suitably normalized[1]. If $\beta$ tends to 0, $P$ tends to the uniform distribution, and implements the single-set strategy of Section 3. As $\beta$ becomes large, a small fraction of the full training set is sampled (99% of the probability mass lies on the $4.6N/\beta$ first elements, 99.99% on the first $9.2N/\beta$). For intermediate values of $\beta$, the model oversamples the first elements in the training set, and undersamples the last: we have a continuous version of two-sample training. To allow for comparison with two-sample training, we define $S_{\text{eff}}$ such that the first $S_{\text{eff}}$ examples in the training set jointly are sampled with probability 25%. In this setting, 10% of the probability mass is on the $0.37S_{\text{eff}}$ first training examples, and 99% on the first $16S_{\text{eff}}$.

For GCD, we experiment with values of $\beta$ ranging from 5.8 to 1152 ($S_{\text{eff}}$ from 25,000 to 5 million)[2]. Table 7 shows that for our training budget of 600 million examples, the best model ($S_{\text{eff}} = 3M$) predicts 65 correct GCD, slightly less than what was achieved with two-set training (Section 4).

Table 7: **GCD for different exponential distributions.** Correctly predicted GCD, best of 5 models, trained on 600 million examples.

| $S_{\text{eff}}$ | 25k | 50k | 100k | 250k | 500k | 1M | 1.5M | 2M | 2.5M | 3M | 3.5M | 4M | 5M |
|---|---|---|---|---|---|---|---|---|---|---|---|---|---|
| $\beta$ | 1152 | 576 | 288 | 115 | 58 | 29 | 19 | 14 | 11.5 | 9.6 | 8.2 | 7.2 | 5.8 |
| GCD | 19 | 21 | 29 | 38 | 46 | 55 | 56 | 57 | 61 | 65 | 63 | 62 | 56 |

For modular multiplication, we need lower $\beta$ (i.e larger $S_{\text{eff}}$) for our training budget of 600M. We report the number of models (out of 25 for each setting) that learn to accuracy above 50% and 95% respectively (Table 8). Again we see that these results are comparable to two-set training (Section 4).

Table 8: **Modular multiplication with different exponential distributions.** 25 models trained on 600 million examples.

| $S_{\text{eff}}$ | 2.5M | 5M | 6M | 8M | 10M | 12M | 14M |
|---|---|---|---|---|---|---|---|
| $\beta$ | 11.5 | 5.8 | 4.8 | 3.6 | 2.9 | 2.4 | 2.1 |
| # Models with 95% accuracy | 2 | 9 | 11 | 13 | 7 | 4 | 3 |
| # Models with 50% accuracy | 4 | 16 | 25 | 22 | 17 | 13 | 6 |

We conclude that the benefits observed in two-set training do not pertain to the specific two-set partition of the training set; rather, it seems that the core of the effect lies in the non-uniform sampling frequency distribution over the (randomly ordered) training set, with a range of frequencies.

### D.5   Varying the optimizer

Some effects observed in deep learning depend on the optimizer, with grokking being a prominent example (Power et al., 2022). Here we provide experimental evidence to show that our findings hold for a variety of optimizers and are thus *robust* and *universal*. We rerun models used for the GCD problem with different optimizers. Specifically, we trained models to predict GCD, with a training budget of 600 million examples, single and two-set training (with $|S| = 50,000$ and $p = 0.25$), and data budgets of 25 million, 50 million and unlimited. We considered four optimizer settings:

- Adam without dropout or weight decay,

---

[1]The normalization factor is $(1 - e^{-\beta})^{-1}$. In our calculations we will approximate it by 1 to simplify computing $S_{\text{eff}}$. For the range of $\beta$ we consider, the resulting approximation error is negligible. In general, for fixed $p$, to compute the size of the set $S(p)$ of first elements that carry probability mass $p$, we can use $\beta \approx -\ln{(1-p)}N/|S(p)|$.

[2]Note that for these values of $\beta$ the distinction between DB 100M and unlimited DB becomes essentially meaningless, as the tails of the training set are sampled exceedingly rarely.

- Adam with weight decay 0.01,
- Adam with dropout (0.1) in the feed-forward networks of the transformer,
- AdamW with weight decay 0.01.

Table 9 presents the best performance of 5 models for each configuration. On average, dropout has an adverse effect on learning, but there is no clear benefit of using weight decay, or AdamW over Adam. Importantly, the separation in performance between single-epoch unlimited training, training on smaller data budgets with more repetitions and two-set training persists across optimizers: the effects we present are robust.

Table 9: **Modular multiplication with different optimizers.** Correctly predicted GCD of the best (of 5) models for various optimizers. The effects we observe are robust under change of optimizer, with a very small degradation for dropout for both the unlimited (single-epoch) and limited DB.

|                  | One-set   |     |     | Two-set   |     |     |
|------------------|-----------|-----|-----|-----------|-----|-----|
|                  | Unlimited | 50M | 25M | Unlimited | 50M | 25M |
| Adam             | 28        | 49  | 61  | 70        | 72  | 63  |
| Adam wd=0.01     | 30        | 56  | 61  | 70        | 70  | 66  |
| AdamW wd=0.01    | 29        | 50  | 58  | 69        | 72  | 67  |
| Adam dropout=0.1 | 24        | 40  | 49  | 66        | 66  | 66  |

