# OpenReview forum: "Repeated examples help learn arithmetic"
_NeurIPS.cc/2024/Workshop/MATH-AI — MATH-AI 24_

### Official Review · Reviewer_1abh · 2024-09-29
**Significant paper with interesting implications**

**Rating:** 8
**Confidence:** 4

**Review:**

1. There are minor typos like "effet" in line 27.
2. What was the motivation behind trying repeated examples? While the paper does show the utility of this approach, I am curious about the genesis of the idea.
3. Could the figures be grayscale-friendly? The bars in Figure 1 are hard to distinguish in grayscale.
4. Line 101 should say "overfitting" instead of "overfit".
5. In Table 1, what is the inflection point of the data budget, for increasing accuracy vs overfitting?

The paper's contribution is quite significant in my opinion.

---

### Official Review · Reviewer_V8Uc · 2024-10-03
**Interesting paper on training small transformers with repeated data**

**Rating:** 7
**Confidence:** 4

**Review:**

Summary: The authors study small transformers trained on GCD and modular multiplication. They find that training on datasets with repeated examples achieve better performance. They also propose a two-set training scheme, where they mix the training set with repetitions of a small random subset of examples.

Strength: The paper is clearly written, and the two-set training scheme is a novel finding.

Weakness: 1. The repetition of a small training set achieves a better performance seems to be an overstatement. The author demonstrates in Fig. 1 that training with a small dataset repeatedly helps model learn faster. However, Fig. 1 also shows that training with larger dataset has not yet converged. It’s likely that once converged, training with a larger dataset may outperform training on the small repeated set. I would suggest the authors rephrasing the claims to emphasize on the speed improvement. I would also guess that the authors’ finding of failure of training on large dataset of modular multiplication task may also be due to the insufficient training time. 2. The paper does not account for how model size might influence the results. For a larger model, maybe training on a larger dataset is faster. I would suggest researching into the dependence on the model size. 3. In Fig. 2, the best performance is achieved at 50 repeated set size, which falls on the left end of the sampling range. This may imply that training without repeated set achieves the best performance. I suggest sampling even smaller repeated set size and include the ones without repeated set size. This also raises the question, why a very small set of repeated set can boost the learning/performance.

Despite the weaknesses mentioned above, I believe the paper offers interesting findings, especially the two-set training scheme that are likely to spark discussion and inspire future work. I believe it will be of interest to many at the workshop.

---

### Official Review · Reviewer_bQJp · 2024-10-06
**Review of "Repeated examples help learn arithmetic"**

**Rating:** 7
**Confidence:** 4

**Review:**

**Summary**

This paper studies the impact of repeating data when learning the greatest common divisor and modular arithmetic problems with encoder-decoder architectures.
The authors first find that repeating a large enough dataset during training can outperform the infinite data regime in both tasks.
They then combine repeating a small dataset whilst also training on a larger infinite dataset, finding that this "two set training" is the most best method in terms of high accuracy for learning these tasks.

**Strengths**
- The study has a strong scientific basis to isolate the specific impact of the dataset changes.
- Provides novel insights on training data mixtures for mathematical data which may apply to other areas in the future.

**Weaknesses**
- The use of bold font seems somewhat random in some parts of the paper, I would suggest using this to bold only key claims. E.g. the bolding of "way worse" on page 3.
- The formatting of the document could overall be improved from larger figure text to using the latex \subsection or \paragraph to structure the document clearly. Also some figures are missing a lot of information to be quickly understood e.g. figure 4 has no y axis label.

**Questions**
- Does this apply to the decoder only case?
- What differentiates this study from the repetition of high quality data within large pretraining datasets? E.g. epoching multiple times on Wikipedia whilst also training on Common Crawl, like is done for Llama models [1]

**An Idea**
- Looking at Figure 1 (right) I see that the models trained on a small set of data repeatedly decrease their test loss fastest. Barring any train/test overlap, it would seem logical to me to try a slightly different variant of "two set training" where the repeated data set decays over time avoiding overfitting but keeping the initial sharp decrease in loss and increase in accuracy. E.g. Start only epoching on the small(er) repeated set then begin to sample from an infinite set decaying over time until all samples are from the infinite regime.


[1] Touvron, H., Lavril, T., Izacard, G., Martinet, X., Lachaux, M.A., Lacroix, T., Rozière, B., Goyal, N., Hambro, E., Azhar, F. and Rodriguez, A., 2023. Llama: Open and efficient foundation language models. arXiv preprint arXiv:2302.13971.

---

### Decision · Program_Chairs · 2024-10-07

Accept